# New System to Increase the Useful Life of Exhausted Barrels in Red Wine Aging

**DOI:** 10.3390/foods9111686

**Published:** 2020-11-18

**Authors:** Francisco Javier Flor-Montalvo, Agustín Sánchez-Toledo Ledesma, Eduardo Martínez Cámara, Emilio Jiménez-Macías, Julio Blanco-Fernández

**Affiliations:** 1Higher School of Engineering and Technology, International University of La Rioja (UNIR), Logroño, 26004 La Rioja, Spain; agustin.sancheztoledo@unir.net; 2Department of Mechanical Engineering, University of La Rioja, Luis de Ulloa 20, Logroño, 26004 La Rioja, Spain; eduardo.martinezc@unirioja.es (E.M.C.); julio.blanco@unirioja.es (J.B.-F.); 3Department of Electrical Engineering, University of La Rioja, Luis de Ulloa 20, Logroño, 26004 La Rioja, Spain; Emilio.jimenez@unirioja.es

**Keywords:** red wine aging, new system of adding wood to wine, barrel oak wood, physicochemical characteristics, oenological parameters, aromatic composition

## Abstract

In recent years, consumers of red wines have demanded aged wines with intense color and a well-integrated fine wood bouquet. Traditionally, wines with these characteristics have been obtained from aging in oak barrels. This type of vinification incurs high costs, including costs associated with the acquisition and use of oak barrels. After five or six vinifications, these barrels are no longer capable of providing an adequate contribution of wood compounds to the wine, because of the exhaustion of their transfer capacity. An alternative to extend the life of these barrels is the introduction of toasted oak staves inside the barrel. In this study, a comparative analysis of the aging of a red wine in new and renewed barrels was developed by inserting toasted staves and analyzing the wine in its different stages, as well as its physical, chemical, and colorimetric characteristics. This study confirms that the use of insert staves anchored in exhausted barrels helps to prolong the useful life of barrels, while maintaining quality assurance.

## 1. Introduction

Wine aging is the maturation process of wine inside a barrel. During this aging process, symbiosis occurs between the tannins in the wine and the tannins in the wood itself, which merge and enrich the wine with their multiple hues and aromas [1,2]. The barrel acts as an oxygenator by means of both the joints between the staves and the porousness of the wood [3,4].

The current trend among wine consumers is moving increasingly toward the consumption of wines with intense color and a well-integrated fine wood bouquet [5,6,7]. In order to produce quality wines with these characteristics, the winemaker’s only possible response is the use of oak barrels in the aging of their wine [3,8,9,10]. This practice, however, gives rise to significant financial costs for wineries, especially due to the high cost of barrels and the fact that their useful life ranges from just two to three aging processes. The replacement of barrels in a winery is therefore a high-cost operation, which in the current economic situation puts a significant financial strain on wineries.

For this reason, over the past few years, there has been a major rise in the use of aging systems that add a trace of wood to wines without the use of barrels, as they are more economical and reduce the final cost of the wine. These systems use different types of oak derivatives: oak chips [11,12,13,14], oak cubes [15], oak powder, shavings, dominoes, or blocks. This practice is common, especially in New World wine-producing countries (Chile, Argentina, South Africa, Australia, and the United States).

It should be noted that these new systems of adding wood to wine are used in order to shorten the wine’s aging time, resulting in the production of wines of a different quality, aimed at a different type of consumer.

In the European Union, this practice was not regulated until the approval of Commission Regulation (EC) No. 1507/2006 on 11 October, 2006, amending Regulations (EC) Nos. 1622/2000, (EC) 884/2001, and 753/2002, which concern certain detailed rules around the implemention of Regulation (EC) No. 1493/1999. The regulation sets out the common organization of the market in the wine industry, regarding the use of pieces of oak wood in winemaking and the designation and presentation of wine so treated. It also sets out the restrictions on and conditions for the use of this material in accordance with the rules approved by the International Organization of Vine and Wine.

In Spain, Royal Decree 1365/2007 applies the provisions of Commission Regulation No. 1507/2006 and updates the description, designation, presentation, and protection of wine products, so that only wines aged in oak containers can be labeled Noble (fine), Añejo (aged), and Roble (oaked). It is also possible to use the traditional complementary terms: Crianza, Reserva, and Gran Reserva.

The decree also specifies that the regulations for quality wines from each region, as well as wines which have a geographical indication encompassing more than one autonomous community, may prohibit the use of oak fragments during wine aging. In the case of Rioja with Qualified Designation of Origin, this prohibition is currently in force under Order APA/3332/2007 of 2 November, 2007, amending the Regulation on Qualified Designation of Origin and its Regulatory Council (Official State Gazette No. 275, 16/11/07).

In this study, we performed a comparative analysis of aging in new oak barrels, along with used and renovated oak barrels, by inserting toasted oak staves that make it possible to reuse the barrels and prolong their useful life. Wine aged in these barrels was analyzed at different stages of aging, in order to determine how aging in new oak barrels can be emulated by this technology at a substantially lower cost. The system tested consists basically of a framework that is placed and anchored inside the barrel. A set of staves with various characteristics is connected within this framework, offering winemakers great versatility in the contribution of wood compounds to the wine during the aging process, through the selection of staves with different toasting levels (light, medium, and medium plus).

## 2. Materials and Methods

### 2.1. Initial Parameters

This research requires the definition of a series of initial parameters, the selection of which determines the results obtained. These initial parameters are:(a)Initial characterization of the wine, grape variety, and type of fermentation. For this study, we used a single-varietal Tempranillo wine [15,16,17]. For the purposes of the study, we decided to follow traditional production methods using malolactic fermentation in barrels [18,19,20].(b)Characterization of the barrels used in testing—type and properties of barrels. All reused barrels were sterilized before their use in the process. Each racking process involved washing the barrels with pressurized water at 80 °C and burning a sulfur stick. In this way, and through the action of SO_2_, contamination was avoided.

The specific data for the base wine can be seen in Table 1 below.

The specific data for the new and reused barrels can be seen in Table 2 below.
(c)Geometry and properties of the staves used for aging the wine. The dimensions of the wood significantly determine its degree of influence on the wine [6]. For this reason, in our study, this task involved designing insert staves of various sizes and geometries, with the aim of identifying the most suitable dimensions for anchoring in an exhausted barrel and allowing the wine to develop as it would in a new barrel. Since there are barrels of various dimensions on the market, the selection of dimensions for the insert staves took into account the model of barrel in which they were to be introduced. Bordeaux barrels were used in this research.

In various studies carried out on the use of alternative systems (powder, chips, cubes, etc.), it has been observed that the release of wood compounds into the wine differs depending on the wood/wine contact surface, with more rapid extraction from smaller-sized systems [21].

Figure 1 shows that insert staves of different geometries and sizes were placed inside the exhausted barrels so that together they occupied 50% of the useful surface of the barrel.

The characteristics of the staves are shown in Table 3.

The tests were conducted on seven different types of barrels. These types are shown in Table 4.

### 2.2. Monitoring the Aging Process

The monitoring of the wine was carried out in two new barrels and five exhausted barrels. The reused barrels are cleaned before use. Staves of various toasting levels were connected inside the barrels [22,23,24]. For the purpose of comparing the results, all the barrels contained the same wine.

The aging of the wine was carried out under controlled temperature and humidity conditions in the winery’s aging area. The wines were aged in a 500 m^2^ facility with no natural light. The conditions maintained within the facility ranged between 90% ± 5% relative humidity and a temperature of 15 °C ± 2 °C. Artificial lighting was provided by halogen lamps and was only used during production-related activities which were necessary for the correct development of the wines, such as removing the barrels from the aging facility for periods of less than 3 h to homogenize the wines in an external stainless steel tank. There was an air recirculation system for standardization of the conditions throughout the facility. The humidification system consisted of a set of water fogging units controlled by an electronic control unit with constant humidity monitoring.

During the aging of the wine the processes of racking, the addition of sulfur, the addition of wine to the barrels because of ullage, etc. were carried out as is considered appropriate for the correct aging of wine under observation.

Throughout the course of the aging, comparative tastings were carried out for each barrel to detect any problems that might have arisen during the wine aging process, such as the appearance of 4-ethyphenol or other undesirable compounds derived from contamination.

Periodically, depending on the winemaker’s criteria, wines aged in the same type of barrels were transferred to a stainless steel tank and blended. Subsequently, the barrels were filled with blended wine from barrels of the same type.

### 2.3. Analysis of Samples

The physical and chemical parameters analyzed, as well as the aromatic composition of the wine as a result of the effects of the wood with which it was in contact, are set out below.

Oenological parameters: pH, °Brix, probable alcohol content, volatile and total acidity, glycerol content, and malic acid content. From the analysis of these parameters it was possible to see the effect of the aging of the wine in barrels containing different types of insert staves on the various oenological parameters. To determine these parameters, we followed the official method set out in Commission Regulation (EEC) No. 2676/90 [25], which has been used by several authors [26,27].Color parameters: color intensity [28], hue, total polyphenol index [29,30,31], anthocyanins [32], and tannins [33,34]. From the analysis of these parameters it was possible to see the effect of the aging of the wine in barrels containing different types of insert staves on the various color parameters.CIELab coordinates: To determine the CIELab coordinates, we followed the methodology of the Commission Internationale de l’Eclairage [35], which have been used by several authors [36,37,38].Volatile compounds from the wood: furfural, 5-methylfurfural, guaiacol, trans-and cis-β-methyl-γ-octalactone (whiskey lactone), eugenol, syringol, and vanillin. The analysis of these compounds allowed us to determine the effect of the various phenolic compounds on the wine from the barrels containing different types of insert staves. To determine the volatile compounds released into the wine from the oak, we used gas chromatography, following the procedure proposed by Ortega et al. [39].

More concretely, the method used to analyze the ellagitannins in the wood samples consisted of maceration in a hydroalcoholic solution followed by liquid–liquid extraction.

The other families of compounds (furan compounds, phenolic aldehydes, phenolic acids, volatile phenols, lactones, and coumarins) were analyzed by accelerated solvent extraction (ASE) with appropriate extraction solvents. This type of extraction is an automated technique that uses high temperature and pressure to carry out extractions quickly and effectively.

Once the relevant compounds had been extracted, the extracts were concentrated under a stream of nitrogen for subsequent detection using chromatographic techniques.

The compounds were classified using high-precision liquid chromatography (HPLC) [40] with a diode array detector (DAD) [41] and gas chromatography (GC) [42] with a mass spectrometry (MS) detector [43].

All the analyses performed for this research were carried out by Fundación Tecnalia Research & Innovation. This service was performed with equipment calibrated and certified by the National Accreditation Body (ENAC). ENAC is the Spanish representative of the European co-operation for Accreditation (EA) and is a signatory of the Multilateral Recognition Arrangement (MLA).

### 2.4. Statistical Analysis

Once the results were obtained, we carried out a statistical treatment using principal component analysis (PCA). PCA allowed us to interpret the similarity in composition between the wooden insert staves and wines analyzed. The wooden insert staves and wines that are near to each other in the figure have a similar composition, whereas those which are distant from each other have a different composition. A vector can be seen in the figure for each compound analyzed. A wooden insert stave or wine that is close to this vector and further from the center of its origin indicates that it has a greater concentration of this compound.

This technique was used to analyze the possible interrelations between:the content of the different aromatic compounds in the aged wine, based on the use of insert staves anchored inside the barrels studied and their toasting level;the impact on the wine’s various physical and chemical parameters during its aging, based on the type of insert staves placed inside the barrels studied.

## 3. Results and Discussion

### 3.1. Aromatic Composition of the Barrels

As stated at the beginning of the article, the aromatization of the wine is one of the main effects sought from aging in oak barrels. While some of the substances transferred to wines are found in natural oak, others only appear after the processes of drying and toasting. With this in mind, sterilization treatments for used oak barrels must ensure not only their microbiological stabilization but must also protect against oxidation and maintain their organoleptic properties.

Table 5 shows the aromatic composition of the used barrels after the cleaning process and therefore the initial conditions for subsequent aging in the barrel. The results for the reused barrel without additional staves are not shown. As can be seen from the results of the analysis of the aromatic composition of the exhausted barrels, the range of concentration detected in the compounds analyzed was considerably lower for the wood analyzed compared with the new barrels. This result is more marked in the family of furanic compounds, volatile phenols, lactones, and tannins.

In this paper we analyze the new wood of barrels B2 and B3, as well as the wood of the insert staves of barrels B4, B5, B6, and B7. This was because the comparative study of the aging of the wines in the different types of barrels, including the wine aged in used barrels, is still of interest in achieving a satisfactory comparative analysis of the development of the wines.

### 3.2. Application in Wine Aging

Table 6 shows the results of the analysis of the aromatic composition of the wine midway through the aging process and after its completion in the barrels.

As can be seen in the table, there is an evolution in the content of the wine’s volatile compounds during its aging in the various barrels studied.

Below is the graph obtained from the statistical analysis.

Figure 2 shows a clear evolution in the wine’s aromatic composition during the aging process in barrels with different types of insert staves anchored inside them. Regardless of the type of barrel in which the wine was aged, after the aging process was complete, it showed a higher content of most of the aromatic compounds, compared to the levels observed midway through the aging process. In the case of the compounds vanillin and syringol, however, their content was lower in all the barrels studied on the completion of the wine aging process compared to wine midway through the aging process, probably due to the evaporation or adsorption of these compounds. If we focus on the different types of barrels, we can see that a different aromatic composition in the wine is achieved depending on the type of insert staves anchored inside each barrel. As can be seen in Figure 2, the wine aged in reused oak barrels containing staves with two types of toasting (B7), both midway through the aging process and on its completion, showed significantly higher content of the compounds syringol and guaiacol compared to the rest of the wines. In contrast, the wine from the standard barrel (B1) and the new barrels (B2 and B3) showed a significantly higher content of the compounds furfural and trans-whiskey lactone after aging, compared to the rest of the wines. At the same time, the wine from the barrels containing staves of a single type (B4, B5, and B6) showed a higher percentage of the compound cis-whiskey lactone. In addition, the wine aged in barrel 4 (B4) showed a significantly higher content of the compound eugenol compared to the rest of the wines. Table 7 shows the results of the analysis of the physical and chemical parameters of the wine during its aging in the barrels.

In order to more clearly visualize the results obtained from the statistical treatment, PCA was performed on the parameters pH, alcohol content, volatile acidity, total acidity, and glycerol, as well as the color intensity parameters A420, A520, A620, hue, CIELab parameters (L*, a*, b*), total polyphenol index (TPI), and total anthocyanins. Malic acid was not considered in the statistical treatment since it was not detected in the wine. Likewise, the parameters °Brix and tannins were not considered, since there was no difference in their values between the wine aged in the different barrels studied.

The results of the analysis (see Table 7) showed no significant difference in the various physical and chemical parameters analyzed between wines aged in the different barrels studied.

Nevertheless, as shown in Figure 3, a different effect can be seen in these parameters depending on the type of insert staves anchored inside each barrel. On the one hand, a higher value can be seen for volatile acidity in the wine aged in the new barrels (B2 and B3) compared to the rest of the wines. On the other hand, barrels B4, B6, and B7 stand out for their higher content of glycerol compared to the rest of the wines. Additionally, the wine aged in barrels B4 and B7 showed a lower pH value compared to the rest of the wines.

Figure 4 shows the different effect on these parameters depending on the type of insert staves anchored inside each barrel. On the one hand, it can be seen that the wine from barrel B3 shows a higher value for color intensity, as well as the yellow component (A420) and the red component (A520), compared to the wine aged in the rest of the barrels. The wine from barrels B6 and B7 showed a higher value for hue compared to the rest of the wines, whereas the wine from barrel B4 showed higher values for the CIELab parameters (L*, a*, and b*). On the other hand, the wine aged in the reused barrel (B1) showed a higher value for TPI with respect to the rest of the wines.

Based on previous studies, it was determined that the concentration of aromatic compounds contributed by the wood fragments increased as the toasting was stronger [44,45], a point that has been contradicted in this study.

With regards to the size of the alternative product, the influence of the contact surface between wine and wood is very important. The use of fragments with a larger contact surface is more similar to the classic oak barrel aging than the use of those with a smaller surface [46].

Through this study, it was determined that the use of staves introduced into the barrel, in a format with a large contact surface (0.116 m^2^), allowed us to emulate classic barrel aging. It is important to remark that the micro-oxygenation surfaces of the barrel were kept identical, as the ratio between the surface of the barrel in contact with the wine and the outside air was not modified in any way, nor was the thickness of the staves.

Several studies confirm that the greater the wood fragment surface area in contact with the wine, the faster the transfer of compounds from the wood to the wine [47].

It is noteworthy that for the contact surface used in this study (1.045 m^2^, approximately equivalent of half a complete barrel, with 2.06 m^2^) and the format of the staves selected, the extraction was slow and similar to that of a new barrel, although in the case of medium and medium plus toasting, the transfer was greater for some representative aromatic compounds.

It is also important to consider the time of contact between wine and wood. In the case of small oak fragments, the extraction of compounds from the wood is very fast, whereas in a new barrel this effect occurs much more slowly [11]. In this study we showed that the extraction of compounds from the wood, and their contribution to the wine in the case of renewing the aromatic capacity of the barrels by adding oak staves, means a slower extraction, closer to the new oak barrels than in the case of using oak fragments with a smaller geometry.

## 4. Conclusions

This study enabled us to determine the effect of the different types of toasting, types of wood, and morphology of implementation, as well as the age of the barrels used. The content of the various aromatic compounds released into the wine during its aging varied depending on the toasting level (untoasted, medium toast, medium toast plus, etc.), with some correlation between this contribution regardless of whether the barrel was used or new, though it was always dependent on the toasting level and source of the wood used.

During the wine aging process, a clear evolution was seen in the wine’s aromatic composition, both with new barrels and with used barrels containing different types of insert staves. This effect, however, was far less noticeable in the case of aging in a used barrel without added insert staves.

Regardless of the type of barrel in which the wine was aged, after the aging process was complete, it showed a higher content of most of the aromatic compounds compared to wine midway through the aging process. The exception to this finding was with regard to vanillin and syringol, where a reduction in their content was observed in all cases studied after completion of the aging process compared to wine midway through the aging process. This phenomenon is mainly due to the evaporation and/or adsorption of these compounds.

The aromatic composition, both during the different phases of the aging process and on its completion, was very different depending on the different types of barrels. Focusing the used barrels with integrated staves, as we can see with the used barrels with light toast staves inside, both midway through the aging process and on its completion, a significantly higher content of the compounds syringol and guaiacol was detected compared to the rest of the wines. The used barrels with medium and medium plus toast staves showed indices of trans-whiskey lactone compared to the rest of the wines, whereas the wine from barrels 4 (B4), 5 (B5), and 7 (B7) showed indices of the compound cis-whiskey lactone.

The highest contents of eugenol after the wine aging process were those from the used barrels with medium toast staves. However, the new barrels with their different toasting levels, after completion of the aging process, showed a significantly higher content of the compounds furfural and trans-whiskey lactone. The new barrels also showed higher levels of volatile acidity and lower concentrations of glycerol than the wines aged in used barrels with or without staves inside.

The color analysis showed that the highest TPI value was obtained from aging in used barrels and especially in those without added staves, whereas with the new barrels, medium toast gave a greater color intensity.

Lastly, it should be noted that the organoleptic evaluation of the wine aged in the barrels showed that the use of different types of insert staves anchored inside exhausted barrels enabled the production of wine with a high-quality bouquet.

The wine industry currently offers wine producers a series of different oak products to be used instead of barrels, consisting of oak fragments with different wood origins, geometries, and toasting levels.

Likewise, there are different possibilities for the use of these products, depending on the moment of application, contact time, dose, etc. All this provides a wide range of possibilities for trying to imitate classic barrel aging.

The results of this study show that staves with different toasting levels, even within the same reused barrel, can offer the winemaker great versatility in terms of the contribution of wood compounds to wine during its aging, by means of the selection of those woods of which the composition (depending on their origin and toasting level) enables the aged wine’s desired bouquet to be achieved.

As a final point, this study confirms that the use of insert staves anchored in exhausted barrels helps to prolong the useful life of barrels, while maintaining quality assurance.

## Figures and Tables

**Figure 1 foods-09-01686-f001:**
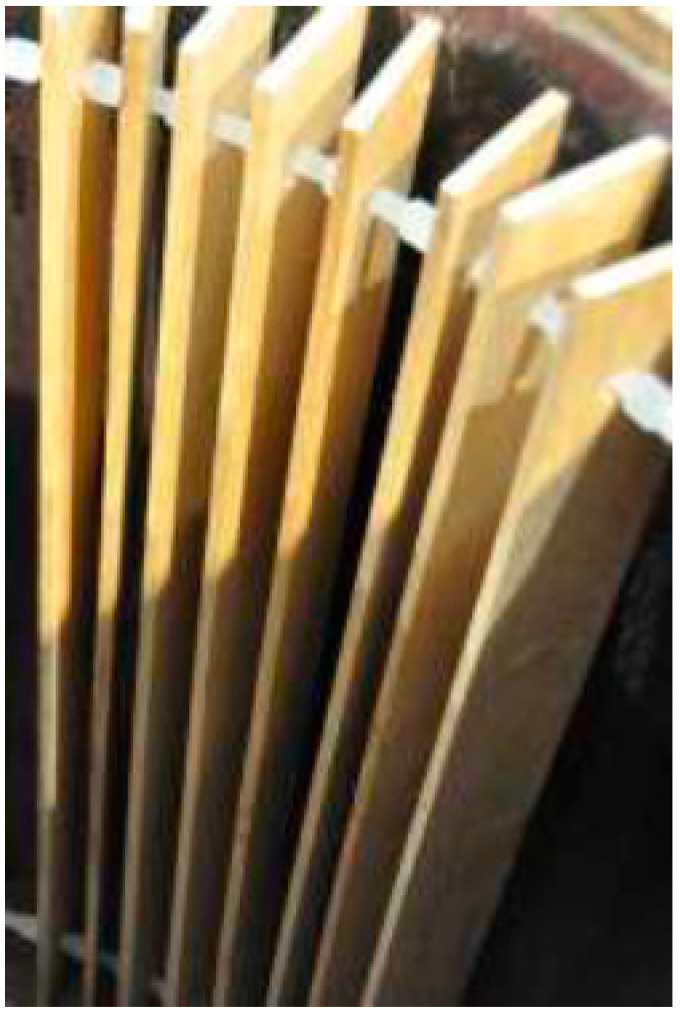
Stave anchoring system designed for use in exhausted barrels.

**Figure 2 foods-09-01686-f002:**
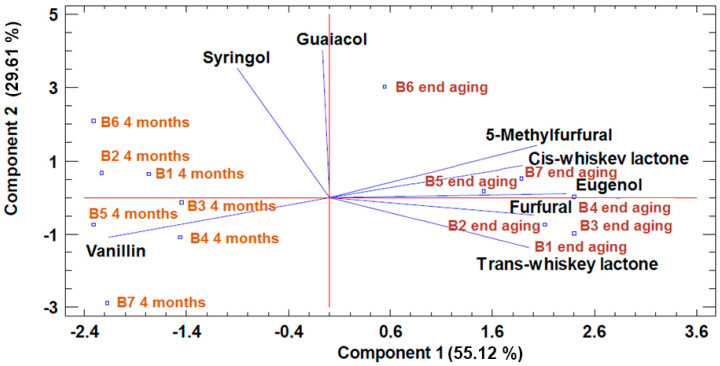
Principal component analysis (PCA) of the aromatic composition of wines aged in the barrels studied.

**Figure 3 foods-09-01686-f003:**
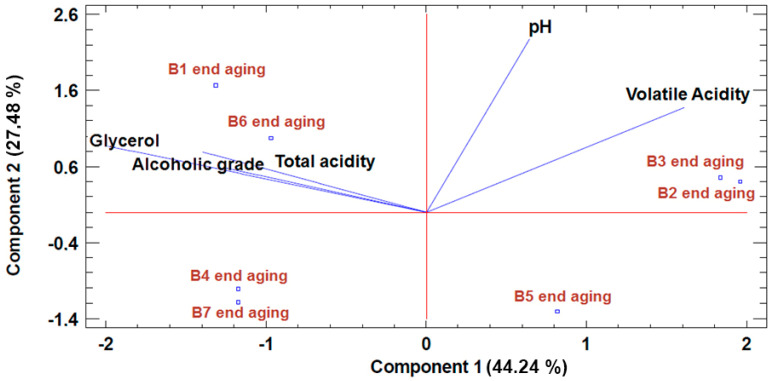
PCA of the parameters pH, alcohol content, volatile acidity, total acidity, and glycerol of the wines aged in the barrels studied.

**Figure 4 foods-09-01686-f004:**
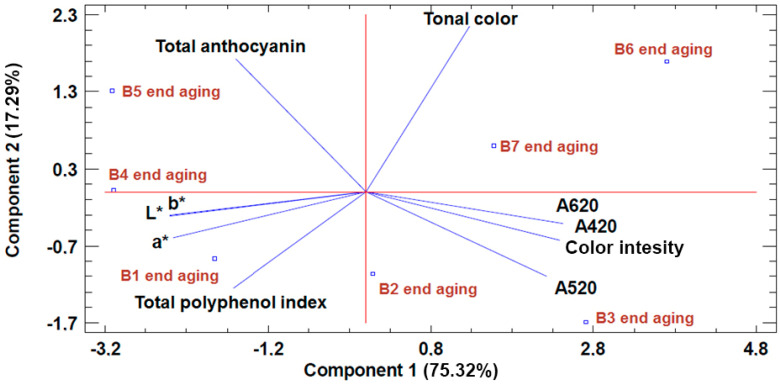
PCA of the color intensity parameters A420, A520, A620, hue, and CIELab parameters (L*, a*, b*). TPI and total anthocyanins of the wines aged in the barrels studied.

**Table 1 foods-09-01686-t001:** Characteristics of base wine.

pH		3.62
Dry extract (g/L)		31.08
Alcohol content (% vol)		13.70
Dry extract (g/L)		31.88
Volatile acidity (g/L acetic acid)		0.49
Total acidity (g/L tartaric acid)		6.30
Malic acid (g/L)		0.17
Tartaric acid (g/L)		3.29
Lactic acid (g/L)		1.53
Sugars (g/L)		2.15
Turbidity (NTU)		48
Total anthocyanins (mg/L)		961
Tannins (g/L)		3.33
Catechins (mg/L)		1718
*Color intensity parameters*		
A420		4.241
A520		11.082
A620		1.970
Color intensity		17.209
Tone		0.388
CIELab parameters	L*	6.34
	a*	34.86
	b*	10.78

**Table 2 foods-09-01686-t002:** Characteristics of used oak barrels.

	New Barrels	Reused Barrels
Model:	Bordeaux Barrels	Bordeaux Barrels
Capacity:	225 + 3 L	225 + 3 L
Wood:	Quercus Alba	Quercus Alba
Thickness of staves:	25–27 mm	25–27 mm
Strapping:	Galvanized steel	Galvanized steel
Barrel stopper:	Food-grade silicone	Food-grade silicone
Contact surface:	2.06 m^2^	2.06 m^2^
Weight:	50 + 5 kg	50 + 5 kg
Years:	New	7 years
Number of vinifications:	-	7

**Table 3 foods-09-01686-t003:** Characteristics of the added staves.

Wood:	*Quercus petraea*
Surface:	0.116 m^2^
Number of staves used:	9
Total contact surface:	1.045 m^2^
Surface equivalent to renew:	50%
Toast:	Light, medium, medium plus

**Table 4 foods-09-01686-t004:** Types of barrels used in the research.

Barrels	Characteristics
B1	Reused barrel. Medium toasted plus and two years of use. It was used as a standard barrel, without introducing staves for aromatic renewal.
B2	New barrel. Medium toasted.
B3	New barrel. Medium toasted plus.
B4	Reused barrel. Nine staves, medium toasted.
B5	Reused barrel. Nine staves, medium toasted plus.
B6	Reused barrel. Nine staves, lightly toasted.
B7	Reused barrel. Nine staves medium toasted and three lightly toasted.

**Table 5 foods-09-01686-t005:** Aromatic composition of the used barrels after the cleaning process.

	B1	B2	B3	B4	B5	B6	B7
Furan compounds (μg/g wood)							
Furfural	34.162 ± 2.977	101.763 ± 8.866	120.388 ± 10.489	55.200 ± 4.809	55.365 ± 4.824	115.885 ± 10.097	196.491 ± 17.119
5-Hethylfurfural	6.023 ± 0.468	5.869 ± 0.456	16.831 ± 1.307	7.170 ± 0.557	8.475 ± 0.658	13.799 ± 1.071	25.941 ± 2.014
5-Hydroxymethylfurfural	3.822 ± 0.169	3.765 ± 0.166	41.998 ± 1.856	4.672 ± 0.206	6.104 ± 0.270	10.644 ± 0.470	71.939 ± 3.179
Phenolic aldehydes (μg/g wood)							
Vanillin	103.445 ± 7.450	144.127 ± 10.380	292.609 ± 21.073	243.622 ± 17.545	341.496 ± 24.594	387.415 ± 27.901	256.308 ± 18.459
Syringaldehyde	587.335 ± 23.323	836.206 ± 33.207	644.510 ± 25.594	742.472 ± 29.484	815.898 ± 32.400	998.149 ± 39.638	713.173 ± 28.321
Phenolic acids (μg/g wood)							
Gallic acid	138.622 ± 9.362	843.693 ± 56.980	391.271 ± 26.425	148.879 ± 10.055	166.266 ± 11.229	131.090 ± 8.853	435.613 ± 29.419
Ellagic acid	2122.202 ± 93.215	3747.298 ± 164.594	3070.444 ± 134.864	2487.237 ± 109.248	2489.367 ± 109.342	2742.838 ± 120.475	2600.524 ± 114.224
Vanillic acid	65.174 ± 2.278	47.428 ± 1.658	68.481 ± 2.393	88.155 ± 3.081	102.070 ± 3.567	203.529 ± 7.113	131.788 ± 4.606
Syringic acid	184.559 ± 13.166	52.590 ± 3.752	93.442 ± 6.666	111.438 ± 7.950	151.917 ± 10.838	314.750 ± 22.454	243.415 ± 17.365
Ferulic acid	<0.322 ± 0.000	<0.320 ± 0.000	<0.353 ± 0.000	<0.322 ± 0.000	<0.319 ± 0.000	<0.319 ± 0.000	<0.318 ± 0.000
Volatile phenols (μg/g wood)							
Eugenol	0.973 ± 0.023	2.225 ± 0.051	1.392 ± 0.032	1.380 ± 0.032	<0.952 ± 0.022	1.253 ± 0.029	3.029 ± 0.069
Syringol	6.359 ± 0.490	5.797 ± 0.447	6.449 ± 0.497	7.729 ± 0.595	6.485 ± 0.500	12.300 ± 0.947	28.615 ± 2.204
Guaiacol	1.007 ± 0.081	0.964 ± 0.078	2.331 ± 0.188	1.273 ± 0.102	1.610 ± 0.130	2.868 ± 0.231	5.281 ± 0.425
4-Methylguaiacol	1.001 ± 0.035	<1.007 ± 0.035	2.469 ± 0.087	1.275 ± 0.045	3.397 ± 0.120	6.925 ± 0.244	4.204 ± 0.148
Lactones (μg/g wood)							
Cis-whiskey lactone	2.006 ± 0.166	6.998 ± 0.578	8.348 ± 0.690	2.241 ± 0.185	2.012 ± 0.166	3.905 ± 0.323	2.690 ± 0.222
Trans-whiskey lactone	1.158 ± 0.108	5.008 ± 0.468	5.601 ± 0.524	1.708 ± 0.160	1.346 ± 0.126	2.104 ± 0.197	1.841 ± 0.172
Coumarins (μg/g wood)							
Aesculetin	<0.395 ± 0.000	<0.397 ± 0.000	<0.438 ± 0.000	<0.400 ± 0.000	<0.397 ± 0.000	<0.397 ± 0.000	<0.395 ± 0.000
Scopoletin	<0.395 ± 0.000	<0.397 ± 0.000	<0.438 ± 0.000	<0.400 ± 0.000	<0.397 ± 0.000	<0.397 ± 0.000	<0.395 ± 0.000
Ellagitannins (μg/g wood)							
Roburin A	14.606 ± 0.444	198.307 ± 6.040	7.644 ± 0.233	11.328 ± 0.345	157.434 ± 4.795	25.119 ± 0.765	104.996 ± 3.198
Roburin B	3.509 ± 0.149	13.894 ± 0.593	1.221 ± 0.052	1.952 ± 0.083	16.042 ± 0.685	4.483 ± 0.191	13.726 ± 0.586
Roburin C	<0.098 ± 0.001	<0.057 ± 0.003	<0.025 ± 0.001	<0.020 ± 0.001	4.635 ± 0.258	<0.242 ± 0.013	<0.613 ± 0.034
Grandinin	8.440 ± 0.441	34.121 ± 1.782	0.744 ± 0.039	0.953 ± 0.050	42.512 ± 2.220	12.185 ± 0.636	34.952 ± 1.825
Roburin D	11.729 ± 0.920	50.836 ± 3.989	1.535 ± 0.120	2.086 ± 0.164	63.032 ± 4.945	9.439 ± 0.741	18.804 ± 1.475
Vescalagina	<0.031 ± 0.002	19.366 ± 1.566	<0.025 ± 0.002	<0.020 ± 0.002	96.114 ± 7.773	<0.395 ± 0.032	<0.613 ± 0.050
Roburin E	6.552 ± 0.269	14.696 ± 0.605	2.995 ± 0.123	3.162 ± 0.130	18.388 ± 0.756	10.411 ± 0.428	32.479 ± 1.336
Castalagin	74.286 ± 6.926	143.904 ± 13.437	23.446 ± 2.189	24.641 ± 2.301	359.505 ± 33.568	119.669 ± 11.174	225.507 ± 21.056

**Table 6 foods-09-01686-t006:** Aromatic composition of the wine midway through the aging process and after its completion in the barrels (Result ± SD).

	Halfway through of Aging Process							End of Aging Process						
	B1	B2	B3	B4	B5	B6	B7	B1	B2	B3	B4	B5	B6	B7
Furfural (μg/L)	524 ± 105	484 ± 97	387 ± 77	526 ± 105	407 ± 81	554 ± 111	160 ± 32	3.439 ± 688	3.202 ± 640	4.696 ± 939	1.398 ± 280	1.094 ± 219	1.124 ± 225	1.754 ± 351
5- Hethylfurfural (μg/L)	216 ± 24	126 ± 14	188 ± 21	121 ± 13	159 ± 17	112 ± 12	75 ± 8	359 ± 39	383 ± 42	659 ± 72	412 ± 45	457 ± 50	646 ± 71	650 ± 72
Guaiacol (μg/L)	21 ± 0	20 ± 0	17 ± 0	15 ± 0	17 ± 0	24 ± 0	10 ± 0	16 ± 0	17 ± 0	17 ± 0	18 ± 0	18 ± 0	24 ± 0	19 ± 0
Trans-whiskey lactone (μg/L)	37 ± 1	49 ± 1	43 ± 1	42 ± 1	49 ± 1	38 ± 1	53 ± 1	136 ± 3	127 ± 3	123 ± 2	86 ± 2	57 ± 1	42 ± 1	72 ± 1
Cis-whiskey lactone (μg/L)	136 ± 5	149 ± 6	204 ± 8	148 ± 6	121 ± 5	142 ± 6	128 ± 5	238 ± 10	271 ± 11	281 ± 11	541 ± 22	454 ± 18	326 ± 13	439 ± 18
Eugenol (μg/L)	17 ± 0	15 ± 0	21 ± 0	17 ± 0	16 ± 0	15 ± 0	15 ± 0	42 ± 1	37 ± 1	36 ± 1	52 ± 1	39 ± 1	30 ± 1	38 ± 1
Syringol (μg/L)	58 ± 1	68 ± 1	57 ± 1	46 ± 1	53 ± 1	79 ± 2	37 ± 1	48 ± 1	47 ± 1	49 ± 1	45 ± 1	44 ± 1	77 ± 2	48 ± 1
Vanillic acid (μg/L)	331 ± 26	368 ± 29	332 ± 27	301 ± 24	412 ± 33	332 ± 27	400 ± 32	226 ± 18	240 ± 19	257 ± 21	256 ± 20	239 ± 19	242 ± 19	269 ± 22

**Table 7 foods-09-01686-t007:** Analysis of the physical and chemical parameters of the wine during its aging in the barrels.

		B1	B2	B3	B4	B5	B6	B7
pH		3.77 ± 0.01	3.76 ± 0.00	3.77 ± 0.00	3.71 ± 0.01	3.73 ± 0.00	3.77 ± 0.00	3.73 ± 0.01
°Brix		8.00 ± 0.00	8.00 ± 0.00	8.00 ± 0.00	8.00 ± 0.00	8.00 ± 0.00	8.00 ± 0.00	8.00 ± 0.00
Alcohol content (% vol)		14.60 ± 0.00	14.60 ± 0.00	14.34 ± 0.37	14.60 ± 0.00	14.34 ± 0.37	14.60 ± 0.00	14.60 ± 0.00
Volatile acidity (g/L Acetic acid)		0.76 ± 0.00	0.82 ± 0.00	0.79 ± 0.00	0.73 ± 0.00	0.73 ± 0.00	0.73 ± 0.00	0.67 ± 0.00
Total acidity (g/L Tartaric acid)		5.4 ± 0.1	5.2 ± 0.0	5.3 ± 0.1	5.3 ± 0.0	5.3 ± 0.0	5.3 ± 0.0	5.3 ± 0.0
Glycerol (g/L)		10.97 ± 0.11	10.83 ± 0.23	10.83 ± 0.21	10.95 ± 0.01	10.84 ± 0.01	10.98 ± 0.08	10.90 ± 0.18
Malic acid (g/L)		<0.25 ± 0.00	<0.25 ± 0.00	<0.25 ± 0.00	<0.25 ± 0.00	<0.25 ± 0.00	<0.25 ± 0.00	<0.25 ± 0.00
Color intensity parameters		12.067 ± 0.010	12.310 ± 0.000	12.732 ± 0.009	11.773 ± 0.001	11.745 ± 0.004	12.610 ± 0.011	12.387 ± 0.001
A420		4.386 ± 0.000	4.471 ± 0.001	4.625 ± 0.002	4.304 ± 0.001	4.289 ± 0.000	4.614 ± 0.002	4.514 ± 0.001
A520		6.018 ± 0.005	6.132 ± 0.001	6.322 ± 0.003	5.860 ± 0.004	5.843 ± 0.001	6.176 ± 0.004	6.118 ± 0.001
A620		1.663 ± 0,004	1.708 ± 0.001	1.785 ± 0.004	1.609 ± 0.004	1.613 ± 0.003	1.821 ± 0.005	1.755 ± 0.001
Hue		0.729 ± 0.001	0.729 ± 0.000	0.731 ± 0.000	0.734 ± 0.000	0.734 ± 0.000	0.747 ± 0.000	0.738 ± 0.000
CIELab parameters	L*	7.965 ± 0.103	7.647 ± 0.014	6.792 ± 0.076	8.749 ± 0.118	8.580 ± 0.076	6.043 ± 0.070	6.659 ± 0.037
	a*	37.752 ± 0.157	37.362 ± 0.025	35.764 ± 0.176	38.900 ± 0.198	38.611 ± 0.124	33.802 ± 0.199	35.285 ± 0.093
	b*	13.700 ± 0.179	13.158 ± 0.025	11.692 ± 0.131	15.042 ± 0.204	14.750 ± 0.131	10.398 ± 0.120	11.457 ± 0.064
TPI - Total Polyphenol Index		61 ± 1	59 ± 1	59 ± 1	60 ± 1	59 ± 0	58 ± 0	59 ± 0
Total anthocyanins (mg/L)		279 ± 7	264 ± 1	254 ± 2	273 ± 5	291 ± 4	271 ± 6	274 ± 1
Tannins (g/L)		3 ± 0	3 ± 0	3 ± 0	3 ± 0	3 ± 0	3 ± 0	3 ± 0

Anova unidimensional analysis was performed for each of these tables with a significance level of α = 0.05. A *p*-value of 1.927 × 10^−17^ was obtained for Table 3, a value 7.7309 × 10^−55^ has been obtained for Table 4, and 1.4157 × 10^−85^ has been obtained for Table 5, from which it can be concluded that all of them are statistically significant.

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
