# Peer review of "New System to Increase the Useful Life of Exhausted Barrels in Red Wine Aging"

_foods, 2020, doi:10.3390/foods9111686_

Round 1
Reviewer 1 Report
This paper provides a method to reuse barrels for wine ageing by inserting a stave anchoring system with several experiments to support their finding. Overall speaking, the paper is easily readable. Here are some major issues I found:
- Table 4 is a big table. How does the conclusion been drawn? Which attribute(s) suggests "the range of concentration detected in the compounds analyzed was considerably lower for the woods analyzed compared with the new barrels." At least I cannot draw the conclusion from the given information.
- B4 is not appeared in figure 2.
- For figure 2, 3 and 4. My understanding is, B2 and B3 are expensive new barrels. Therefore, their readings should be the goal of this research. In other words, If B4~B7 can give similar results to B2 and B3, that can justify we can use cheaper way to make similar wines with new Barrels. However, in figure 2~4, the standing of B4~7 are usually very far from B2 and B3. This means the wines made from new method are DIFFERRENT from new barrels. Are the wine made from the proposed approach is better or equal to wine made from new barrels?
- Similar to the previous point, in line 211, “after the ageing process was complete, it showed a higher content of 211 most of the aromatic compounds with respect to wine midway through the ageing process”; is HIGHER better? How can this point been justified. Maybe it’s overpowered.
- For table 6, Many of these tests are about the wine itself, since the wine in all Barrels are the same, I am not surprise to see the values listed in this table are similar. How does wines taste might be more important. Is it possible to have some wine reviewers to evaluate the final product of the wine to indicate that the wines aged in the proposed approach are indeed similar or even better?
Minor corrections:
- Throughout the paper m2 should be
- Line 84, a single-varietal Tempranillo wine [16,17]. Some words are bold others are not, which may not be necessary.
- Table 2,
|
Surface: |
0,116 m2 |
I cannot understand 0,116 as a value
Author Response
Table 4 is a big table. How does the conclusion been drawn? Which attribute(s) suggests "the range of concentration detected in the compounds analyzed was considerably lower for the woods analyzed compared with the new barrels." At least I cannot draw the conclusion from the given information.
Thank you very much for this comment. Attending to your appreciation, an additional column has been included corresponding to the barrel used without renovation by adding new wood to facilitate the analysis. We consider that the comparison of this barrel with the new barrels and the used barrels with wood cession compounds renovation by insertion of staves is now much clearer.
In this way, the main aromatic compounds that affect the taste and aroma of wines, such as furfural (almond and toasted almond aromas), vanillin (vanilla aromas),..., have considerably lower levels in new barrels than in used ones.
B4 is not appeared in figure 2.
Thank you very much for this comment. Attending to your appreciation, we have corrected figure 2 by including sample B4.
For figure 2, 3 and 4. My understanding is, B2 and B3 are expensive new barrels. Therefore, their readings should be the goal of this research. In other words, If B4~B7 can give similar results to B2 and B3, that can justify we can use cheaper way to make similar wines with new Barrels. However, in figure 2~4, the standing of B4~7 are usually very far from B2 and B3. This means the wines made from new method are DIFFERRENT from new barrels. Are the wine made from the proposed approach is better or equal to wine made from new barrels?
Thank you very much for this comment.
In wines aging with oak barrels, the influence of the type of wood and its toasting is very important. More specifically, from the same oak wood, depending on the times and temperatures of toasting of that wood, the levels of furanic compounds (especially furfual), phenolic compounds (with special influence on the contribution of vanillin and ellagic acid) or volatile phenols (especially eugenol, syringol and guaiacol) will change substantially.
As a barrel is used in successive vinifications, its capacity to yield these compounds to the wine is reduced by exhaustion of these compounds. For this reason, this study proposes to "renew" the barrels' capacity to yield aromas by inserting toasted staves inside them.
Obviously, depending on the typology, origin, toasting and total contact surface of the staves, we will be able to achieve various aromatic transfers, allowing us to adapt the new aromatic transfer to the tastes of the wine maker.
In any case, other advantages of the barrel vinification remain, mainly derived from the micro-oxygenation during the aging.
In view of the above, it is not possible nor is it the object of this study to determine whether wines obtained from renewed barrels with inserted staves are better or worse than those aged in new barrels, given that this parameter is not quantifiable for us.
However, we can affirm that the use of renewed barrels with insertion of toasted oak staves allows us to compensate for the transfer of aromatic compounds and we can deduce that, by adjusting the contact surface, wood origin and toasting in the contribution of these staves, we can achieve results similar to those obtained in new barrels.
We proceed to modify the conclusions section to clarify this point.
Similar to the previous point, in line 211, "after the ageing process was complete, it showed a higher content of 211 most of the aromatic compounds with respect to wine midway through the ageing process"; is HIGHER better? How can this point been justified. Maybe it's overpowred.
Thank you very much for this comment.
As we explained in the answer to the question above, the use of barrels, after several vinifications, has the problem of considerably reduction for the contribution of aromatic compounds to the wine. This study seeks to contrast the capacity of the barrel renovation system by inserting toasted staves in order to recover that capacity for aromatic release.
Although a greater aromatic transfer is not necessarily positive (it will depend on what the winemaker is looking for through the aging process), the aging in aromatic exhausted barrels has no reason to be used, since it will only provide a micro-oxygenation together with a minimum aromatic contribution. That is why the use of barrels with more than 5 complete vinifications is not recommended.
As an example, winemakers tend to mix different barrels (both in terms of age, toasting and wood origin) in order to obtain a wine. Sometimes, they also mix different varietals of base grapes) to later join them together and obtain the appropriate mixture which they call "coupage".
As you can see, in these circumstances, being able to have renewed barrels would provide an adaptable tool for barrel aging (since different numbers of staves, different geometries, toasting, ...) while the use of used barrels, allows virtually no contribution.
We proceed to modify the conclusions section to clarify this point.
For table 6, Many of these tests are about the wine itself, since the wine in all Barrels are the same, I am not surprise to see the values listed in this table are similar. How does wines taste might be more important. Is it possible to have some wine reviewers to evaluate the final product of the wine to indicate that the wines aged in the proposed approach are indeed similar or even better?
Thank you very much for this comment.
As you said, the physical parameters in table 6 must be similar since they are the same base wine.
However, the differences in color are remarkable.
The differences in the taste and smell of the wines do not depend on these parameters, although they are important in determining the possibilities of evolution of the wines. For this reason, we thought it important to indicate them in the study (as an example, the alcoholic content directly determines the capacity of a wine to absorb aromatic compounds from the wood).
As you suggest, it would be interesting for experts to carry out a blind tasting of the wines and determine the suitability of the wood mixtures used for the inserts in the renovated barrels.
We proceed to include this point into futures actions for this line of investigation.
Throughout the paper m2 should be m2.
Thank you very much for this comment. We have corrected this aspect throughout the article.
Line 84, a single-varietal Tempranillo wine [16,17]. Some words are bold others are not, which may not be necessary.
Thank you very much for this comment. We proceed to replace the term tempranillo monovarietal by tempranillo wine.
Table 2, Surface: 0,116 m2. I cannot understand 0,116 as a value
Thank you very much for this comment. We have include m2 into table 2 for surface data.
Reviewer 2 Report
The manuscript reports an investigation about the application of a novel and low cost technical system to prolong the life of oak barrels. The Authors claimed that the use of insert staves internally fixed in spent barrels helped to extend the active life of barrels.
The paper deals with a topic of general interest and, indeed, it describes for the first time the above described approach. However, before publication some major points needs to be clarified as outlined below:
Lines 83-84. The chemical and physical properties of the Tempranillo wine employed must be indicated.
Line 139. The methods used for the determination of volatile molecules, phenolic acids and coumarins must be reported. In particular for phenolic acids and coumarins there are no references to extraction methods while for volatiles determination the authors used a methods developed by Ortega et al. (2001), on other classes of volatile molecules, so it is important to indicate in particular any changes made to the method and the chromatographic conditions used to identify volatiles derived from barrels.
Lines 184-185 and lines 190-192. I do not see the scientific meaning of the reasoning that it is not necessary to show the results of reusing the barrels without adding staves. These data are important and they must be included. Moreover, they have been statistically analysed by the Authors as reported in Figs 2 and 3.
Lines 239-244. How do the Authors explain the results shown in Fig 3 considering the data presented in the Table 6? For example, how B4 and B5, that show almost identical pH. alcohol content. volatile acidity. total acidity and glycerol values, could lay on different areas of the PCA graphical output?
Figures 2-3-4. Since the percentages of the principal components have not been indicated in the figures, it is not possible to evaluate the significance of the analysis. They must be indicated.
Table 3. Specify the significance of the symbol “+” in B1, B3 and B5 descriptions
Tables 4-5-6. A one-way ANOVA analysis needs to be carried out, indicating in the table the consequent grouping.
The Results and Discussion section needs to be remodulated. In fact, it just consists of a description of the obtained results. A proper discussion of the presented data and their comparison with those described in similar investigation is totally missing.
Author Response
Lines 83-84. The chemical and physical properties of the Tempranillo wine employed must be indicated.
Thank you very much for this comment. Attending to your appreciation, we have proceeded to include a new table (Table 1) in which the physical-chemical characteristics of the base wine are exposed.
Line 139. The methods used for the determination of volatile molecules, phenolic acids and coumarins must be reported. In particular for phenolic acids and coumarins there are no references to extraction methods while for volatiles determination the authors used a methods developed by Ortega et al. (2001), on other classes of volatile molecules, so it is important to indicate in particular any changes made to the method and the chromatographic conditions used to identify volatiles derived from barrels.
Thank you very much for this comment. Attending to your appreciation, we have described the methodologies used for the determination of the volatile components. Such description can be found in lines 156 to 166.
Lines 184-185 and lines 190-192. I do not see the scientific meaning of the reasoning that it is not necessary to show the results of reusing the barrels without adding staves. These data are important and they must be included. Moreover, they have been statistically analysed by the Authors as reported in Figs 2 and 3.
Thank you very much for this comment. Attending to your appreciation, we have included a new column in table 4, which includes all the characteristics of the wine in the used barrel without bringing in new oak staves.
Lines 239-244. How do the Authors explain the results shown in Fig 3 considering the data presented in the Table 6? For example, how B4 and B5, that show almost identical pH. alcohol content. volatile acidity. total acidity and glycerol values, could lay on different areas of the PCA graphical output?
Thank you very much for this comment. Attending to your appreciation, we proceed to explain the above point.
A study of the analytical results associated with the samples B4 and B5 shows differences that, although apparently (in absolute value) seem to be small, they are relatively very large, analysed in the range of each magnitude.
More explicitly, the values of Alcoholic Content (14.60 % vol and 14.34 % vol respectively for samples B4 and B5) and Glycerol (10.95 g/l and 10.84 g/l respectively for samples B4 and B5) are at the extremes of the range obtained for the total of the samples (10.34 % vol to 14.60% vol for alcoholic content and 10.83 g/l to 10.95 g/l for Glycerol).
These two parameters are both of alcoholic chemical origin and they are related. That is why, when projecting each of the parameters, samples B4 and B5 are very close on the vertical axis, while they are far away on the horizontal axis. This reveals their similarity in terms of pH and acidity and their important differences in terms of alcoholic compound content.
Figures 2-3-4. Since the percentages of the principal components have not been indicated in the figures, it is not possible to evaluate the significance of the analysis. They must be indicated.
Thank you very much for this comment. Attending to your appreciation, we have included the % for the principal components in Figures 2, 3, and 4.
Table 3. Specify the significance of the symbol “+” in B1, B3 and B5 descriptions.
Thank you very much for this comment. Medium + toasting is stronger than traditional medium toasting. It is the usual name used by manufacturers for this type of toasting. In order to avoid confusion the designation medium + is replaced by medium plus throughout the article.
Tables 4-5-6. A one-way ANOVA analysis needs to be carried out, indicating in the table the consequent grouping.
Thank you very much for this comment. Attending to your appreciation, we have performed a one-way ANOVA analysis for each of the tables and the results are presented in the paper.
The Results and Discussion section needs to be remodulated. In fact, it just consists of a description of the obtained results. A proper discussion of the presented data and their comparison with those described in similar investigation is totally missing.
Thank you very much for this comment. Attending to your appreciation we have proceeded to extend such section including a real discussion, which effectively was incomplete in the previous version.
Round 2
Reviewer 1 Report
I believe the authors have answered all my questions and made the corresponding changes.